# Evaluation of Pharmacokinetics and Pharmacodynamics of Deferasirox in Pediatric Patients

**DOI:** 10.3390/pharmaceutics13081238

**Published:** 2021-08-11

**Authors:** Laura Galeotti, Francesco Ceccherini, Carmen Fucile, Valeria Marini, Antonello Di Paolo, Natalia Maximova, Francesca Mattioli

**Affiliations:** 1Phymtech Srl, Via F.lli Rosselli 8, San Giuliano Terme, 56127 Pisa, Italy; laura.galeotti@phymtech.com (L.G.); francesco.ceccherini@phymtech.com (F.C.); 2Department of Internal Medicine, Pharmacology & Toxicology Unit, University of Genoa, 16100 Genoa, Italy; antidpx@gmail.com (C.F.); valeria.marini@unige.it (V.M.); Francesca.mattioli@unige.it (F.M.); 3Clinical Pharmacology Unit, EO Ospedali Galliera, 16128 Genoa, Italy; 4Department of Clinical and Experimental Medicine, Section of Pharmacology, University of Pisa, Via Roma, 55, 56126 Pisa, Italy; 5IRCCS Burlo Garofolo, Bone Marrow Transplant Unit, Institute for Maternal and Child Health, 34137 Trieste, Italy; nataliza.maximova@burlo.it

**Keywords:** Deferasirox, children, population pharmacokinetics, tolerability, therapeutic drug monitoring

## Abstract

Background: Deferasirox (DFX) is commonly used to reduce the chronic iron overload (IO) in pediatric patients. However, the drug is characterized by a large pharmacokinetic variability and approximately 10% of patients may discontinue the treatment due to toxicities. Therefore, the present retrospective study investigated possible correlations between DFX pharmacokinetics and drug-associated toxicities in 39 children (26 males), aged 2–17 years, who underwent an allogeneic hematopoietic stem cell transplantation. Methods: IO was diagnosed by an abdominal magnetic resonance imaging and DFX was started at a median dose of 500 mg/day. DFX plasma concentrations were measured by a high performance liquid chromatographic method with UV detection and they were analysed by nonlinear mixed-effects modeling. Results: The pharmacometric analysis demonstrated that DFX pharmacokinetics were significantly influenced by lean body mass (bioavailability and absorption constant), body weight (volume of distribution), alanine and aspartate transaminases, direct bilirubin, and serum creatinine (clearance). Predicted DFX minimum plasma concentrations (C_trough_) accounted for 32.4 ± 23.2 mg/L (mean ± SD), and they were significantly correlated with hepatic/renal and hematological toxicities (*p*-value < 0.0001, T-test and Fisher’s exact tests) when C_trough_ threshold values of 7.0 and 11.5 mg/L were chosen, respectively. Conclusions: The population pharmacokinetic model described the interindividual variability and identified C_trough_ threshold values that were predictive of hepatic/renal and hematological toxicities associated with DFX.

## 1. Introduction

Chronic iron overload (IO) is a serious consequence of blood transfusions in patients affected by myelodysplastic syndromes, thalassemia, and sickle cell disease, regardless of age [1]. In patients affected by hematological malignancies or who underwent a hematopoietic stem cell transplantation (HSCT), IO is considered multifactorial [2]. Intensive cytotoxic therapy before HSCT causes bone marrow and neoplastic cell lysis, releasing free and protein-bound iron and inducing excessive iron storage [3]. IO has been associated with poor prognosis in patients undergoing allogeneic HSCT, correlating with an increased risk of non-relapse mortality and acute and chronic graft-versus-host disease (GVHD) [4,5,6]. The severity of siderosis depends on the progressive damage of tissues and organs (i.e., liver, kidney, heart, and endocrine glands) through the formation of free oxygen radicals [7,8,9]. The advent of specific iron chelators such as deferasirox (DFX) has represented an effective treatment to lessen the iron content in the body and to prevent the subsequent tissue damage. Although the drug has a good tolerability, DFX is characterized by non-negligible risks and toxicities that may require the temporary discontinuation of drug administration or other supportive therapies [10]. In pediatric patients, gastrointestinal disturbances, as well as diarrhea, nausea, vomiting, and liver toxicities are common adverse events, with an incidence of 1–10%. Instead, the increase in serum creatinine occurs more frequently (30–100% of patients), depending on the dose and on the enrolled population [11]. In particular, drug-induced liver injuries (DILI) in children are less frequent than in adults, but they remain a serious concern [12]. Overall, adverse reactions to DFX are likely responsible for approximately 10% of drug discontinuations in both children and adults [10].

A highly variable pharmacokinetic profile among patients significantly contributes to the different degree of tolerability of the drug. Indeed, DFX is rapidly absorbed through the gut but the final bioavailability depends on the meal eaten [11]. Moreover, DFX is highly bound to plasma proteins (>99%), and it undergoes to hepatic biotransformation through glucuronidation (CYP biotransformation is a minor route of metabolism), with the following excretion into the feces as parent drug (60%) and metabolites (8.3%) [13]. It is worth noting that pediatric patients display a lower exposure (from −20% up to −30%) when compared to adults [14]. Although the initial dose should be the same in children and adults, age-related adjustments should be taken into consideration in pediatric populations. Therefore, because of (i) the intrinsic physiological characteristics of the pediatric population, (ii) the changes in drug pharmacokinetics across different ages, and (iii) the interindividual variability in tolerability, the adoption of a therapeutic monitoring protocol of DFX may be recommended [15,16,17].

The aim of the present study is two-fold. First of all, to build a population pharmacokinetic (POP/PK) model to evaluate the drug concentration over time in every patient and hence to predict the individual dose value in terms of the patient’s covariates. Secondly, to verify the C_trough_ threshold and determine whether such a criterion applies to every type of toxicity associated with DFX or not. The cohort of patients is represented by the pediatric population of allogeneic HSCT recipients affected by severe systemic siderosis and undergoing chelation therapy with DFX.

## 2. Materials and Methods

### 2.1. Patients

All pediatric patients (aged 2–17 years) who received DFX at the Bone Marrow Transplant Unit, IRCCS Burlo Garofolo, Trieste, Italy, were selected to participate in the present retrospective study. All study patients had undergone allogeneic HSCT preceded by a myeloablative conditioning regimen [18].

During the pre-transplantation work-up, all patients had undergone an abdominal magnetic resonance imaging (MRI)-based evaluation of iron concentration in the liver, pancreas, spleen, and bone, as previously described [2]. All patients with severe systemic iron IO received chelation therapy with DFX in the post-transplant period.

After the informed consent was signed by the parents or a legal guardian, data of interest (i.e., age, weight, clinical chemistry values, therapies, therapeutic drug monitoring values, etc.) were obtained in a complete anonymized fashion to protect patients’ privacy. In particular, the harvested data included measures for multiple occasions of DFX measurements of plasma levels at steady state. The planning and the execution of this study were approved by the Ethics Committee of the Institute for Maternal and Child Health, IRCCS Burlo Garofolo (Reference no. 1105/2015, ClinicalTrials.gov identifier: NCT04423237).

### 2.2. Blood Withdrawal and Measurement of Deferasirox Plasma Concentrations

In the whole population of enrolled patients, blood samples to measure DFX concentrations were harvested during routine monthly visits, immediately before the daily drug administration (C_trough_) when the steady state was achieved. In 7 patients, additional blood samples were collected during a dense sampling scheme between two consecutive administrations at steady state to capture the complete plasma profile of the drug, which in turn enabled the efficient elaboration of the pharmacokinetic model. In particular, the 8 samples were collected over the entire 24-h time interval, at precisely 1, 2, 4, 6, 9, 12, and 22 h after the intake of the DFX daily dose. The latter sample was collected immediately before the following dose (i.e., the 24-h sample). For every sample, the exact time of blood withdrawal was recorded in the study database. Each blood sample (4 mL) was collected into heparinized Vacutainer© tubes, and plasma resulting from centrifugation at 1000× *g* for 10 min was stored at −20 °C until the analysis.

The measurement of DFX plasma concentrations was performed by adopting a validated high-performance liquid chromatographic (HPLC) method with UV detection [19] within a routine protocol of therapeutic drug monitoring (TDM). Briefly, plasma proteins were precipitated by methanol, then the sample was diluted with water. A 100-μL aliquot was injected within the HPLC system (Ultimate 3000 HPLC system, ThermoFisher Scientific Inc., Waltham, MA, USA) and isocratically eluted by mobile phase constituted by buffer (0.05 M Na_2_HPO_4_ and 0.01 M tetrabutylammonium hydrogen sulfate), acetonitrile and methanol (42:12:46, *v*/*v*/*v*). Separation of peaks of interest was done by using an Alltech Alltima C_18_ column (250 × 4.6 mm, 5 μm) and absorbance was measured at 295 nm. The method was proved to be reliable as it showed coefficients of intra- and inter-day variability lower than 15%. Furthermore, limits of quantitation (0.5 mg/L), and the range of linearity (0.5–60 mg/L) enabled the correct measurement of DFX plasma concentrations [20].

### 2.3. Pharmacokinetic Analysis

The disposition of DFX was studied through a population pharmacokinetic (POP/PK) analysis utilizing a one-compartment POP/PK model with absorption and employing a nonlinear mixed effect modelling approach (NONMEM 7.3^®^, ICON, Dublin, Ireland). The development of the pharmacokinetic model was limited to a single compartment approach because of the high variability encountered in the pharmacokinetic curves of the different patients. The developed model, although not able to catch all the fine details, is nevertheless sound and robust enough to describe most of the underlying dynamics that deserved interest for our clinical purposes.

Notably, the individual case report forms returned several variables that could influence the pharmacokinetics of DFX. Therefore, the interaction plot analysis and the factor analysis of mixed data (FAMD) guided a preliminary selection of the covariates for their possible inclusion into the model. The main goal of the FAMD technique is to detect covariates or aggregations of covariates that are potentially informative in differentiating the population. Indeed, FAMD allows one to distinguish different so-called “dimensions” that, combined together, represent the population total variance. The specific contribution of a particular dimension to the total variance is a key parameter utilized to assess the role of that dimension. Similarly, within each dimension, each covariate is analyzed to evaluate its role in that specific dimension. As a rule of thumb, in the development of a new pharmacokinetic model, covariates being significant in the most significant dimensions are expected to suggest a possible covariate aggregation, but no indication is provided in terms of the type of dependence or of the level of complexity. Therefore, the variables were selected and taken into consideration in the development of the model if they represented a significant part of the DFX pharmacokinetics variability. The effect of covariates on pharmacokinetic parameters was tested using linear and nonlinear (i.e., power or piece-wise) correlations.

The decrease in objective function value (OFV) and goodness-of-fit plots guided the development of the model, together with bootstrap analyses and visual predictive checks (VPC) through the use of the PsN toolkit and the Xpose package [21,22,23]. Perl and R (release 3.3.3) environments were used for these purposes.

For each patient, the predicted C_trough_ value was compared with the measured value and subsequently used for the statistical analyses between groups according to the registered toxic effects.

### 2.4. Toxicity Criterion

Often, patients exhibit different kinds of adverse events simultaneously. Therefore, we decided to verify whether we could assess specific correlations between the C_trough_ value and each particular category of adverse events rather than considering all kinds of toxicities as a single *ensemble*. In particular, we identified three different groups of toxic events: (A) gastrointestinal, (B) hematological, and (C) hepatic/renal events. Patients exhibiting hepatic and renal adverse events were included in the same group because according to our data all patients who showed renal discomforts showed hepatic adverse events as well. The breakdown of patients who experienced toxicities was as follows: 7 for group A, 16 for group B, and 18 for group C. Every possible correlation between C_trough_ and a particular category of adverse events has been assessed by the T- and Fisher′s exact tests.

In order to verify the existence of a possible cut-off value for C_trough_, a receive operating characteristics (ROC) analysis was performed. The Youden Index was adopted for the identification of the most appropriate cut-off value. When the cut-off value was identified, the T-test was used to compare the means of C_trough_ values for patients who showed a particular toxicity versus the corresponding values for patients who did not show that toxicity. In addition, the Fisher’s test was performed, taking into account how many patients who exhibited or did not exhibit the toxicity under consideration had C_trough_ value higher or lower than the cut-off value.

### 2.5. Statistical Analyses

We expressed continuous variables through the mean value (standard deviation, SD) for normally distributed variables and through the median value (interquartile range, IQR) for the non-normally distributed variables. Qualitative variables were expressed through frequencies and percentages. Statistical checks to assess differences among mean values in different groups have been performed through a T-test for two groups and through a one-way analysis of variance (ANOVA) with post-hoc Tukey HSD (honestly significant difference) test for multiple group comparisons [24]. As an additional verification of the statistical strength of the final results, statistical analyses included Scheffé, Bonferroni, and Holm calculations for multiple comparison analysis.

## 3. Results

### 3.1. Patients

The present pediatric population of patients included 39 children (M/F ratio, 26/13), aged 2–17 years, who received a median DFX dose of 500 mg/day (range, 406–1000 mg/day) as summarized in Table 1. The results of TDM protocol returned a median value of minimum plasma concentrations of 32.4 ± 23.2 mg/L, without significant differences (*p* = 0.131) between male (20.9 ± 8.3 mg/L) and female patients (41.9 ± 28.1 mg/L).

### 3.2. FAMD Results

The first two dimensions identified by the FAMD technique represent a cumulative variance higher than 50%, namely 29.4% and 21%, respectively. Hence, it was considered appropriate to limit our analysis to these two dimensions only. Figure 1 shows the most significant covariates from Dimension 1 (age, weight, lean body mass, and creatinine) and Dimension 2 (ALT, direct bilirubin, and AST).

### 3.3. POP/PK Modeling

The elaboration of a POP/PK model was completed utilizing a one-compartment model with mixed error, and it was performed using the NONMEM’s *advan2* and *trans2* methods. The obtained final parameterizations of bioavailability (F_1_), absorption rate (k_a_), clearance (Cl), and volume of distribution (V) are (Equations (1)–(4)):(1)$$F_{1}=0.7-1.2\;{\left(\frac{LBM}{60}-0.5\right)}^{2}$$
(2)$$\begin{array}{c} Cl=\theta_{1}e^{-\left(\theta_{4}\times \frac{Bil_{dir}}{Bil_{dir}{}_{th}}\times \frac{ALT}{ALT_{th}}\times \frac{Crea}{Crea_{th}}\right)}e^{\eta 1}\;\\ \mathrm{if}\;\left(\frac{Bil\_dir}{Bil\_dir_{th}}<1,\;\frac{ALT}{ALT_{th}}<1,\frac{Crea}{Crea_{th}}<1,\frac{AST}{AST_{th}}<1\right)\\ \mathrm{else}\\ Cl=\theta_{1}e^{-\left(3\theta_{4}\times \frac{Bil\_dir}{Bil\_dir_{th}}\times \frac{ALT}{ALT_{th}}\times \frac{Crea}{Crea_{th}}\right)}e^{\eta 1} \end{array}$$
(3)$$V=\theta_{2}WT^{0.75}e^{\eta 2}$$
(4)$$k_{a}=\theta_{3}e^{-\left(\frac{LBM}{60}\right)}e^{\eta 3}$$
where *LBM* and *WT* are lean body mass and body weight, respectively, while *Bil_dir_th_*, *ALT_th_*, and *AST_th_* are the thresholds of the laboratory exams of direct bilirubin (0.2 mg/dL), alanine transaminase (45 U/L), and aspartate transaminase (45 U/L), respectively. On the contrary, the threshold value for creatinine (*Crea_th_*) had been taken into account as dependent on the gender and on the age of the patient as showed in Table 2. This choice was motivated by the fact that the kidney functionality is characterized by a wide variability across different age values [24].

Table 3 summarizes the results of the PK final model, together with the findings of the bootstrap analysis. The goodness of fit plots and the visual predictive checks are shown in Figure 2 and Figure 3, respectively.

Finally, the findings regarding the correlation between predicted and measured C_trough_ values showed that a highly significant linear correlation was achieved (Figure 4).

### 3.4. Correlations between Plasma Concentrations of DFX and Toxicities

From the pharmacokinetic model we computed the predicted C_trough_ for all patients and occasions, and we simultaneously compared the mean value of four different C_trough_ groups given by (A) measured C_trough_ value for patients with hepatic/renal toxicity; (B) predicted C_trough_ value for patients with hepatic/renal toxicity; (C) measured C_trough_ value for patients without hepatic/renal toxicity, and (D) predicted C_trough_ value for patients without hepatic/renal toxicity. The *p*-value corresponding to the F-statistic of the one-way ANOVA performed on these four groups was considerably lower than the 0.05 significance threshold, namely, *p*-value = 3.9 × 10^−15^. Hence, the difference between the groups was statistically significant according to the final results of the ANOVA test (Appendix A).

To identify which of the pairs of groups were significantly different from each other and to take into account the multiple comparison problem, we performed four additional multiple comparison tests (post-hoc Tukey HSD, Scheffé, Bonferroni, and Holm tests) that provided aligned results (Appendix A). Indeed, all multiple comparison tests provided the same results, i.e., the differences in the mean values of C_trough_ in groups A and B, as well as the differences in the mean values of C_trough_ in groups C and D, were not statistically distinguishable. On the contrary, the mean values of C_trough_ for groups A vs. C, B vs. C, A vs. D, and B vs. D were statistically different with a *p*-value < 0.01. Therefore, the comparison between group A and B showed that the measured and PK-predicted C_trough_ values in patients with renal/hepatic toxicity were not different, hence confirming the validity and quality of the model. A similar result has been obtained for the measured and predicted C_through_ mean values in patients that did not show hepatic/renal toxic events (groups C vs. D). The other four comparisons demonstrated that the mean values of C_trough_ for patients that exhibit hepatic/renal toxicity and for patients not experiencing renal/hepatic toxicity are always statistically different no matter whether the values are either measured or predicted by the pharmacokinetic model.

Finally, the same analysis was applied to hematological toxicities, resulting in findings that were superimposable to those obtained for hepatic/renal toxicities (Appendix A).

### 3.5. Identification of the C_trough_ Threshold

The ROC analysis of predicted C_trough_ values according to hepatic/renal and hematological toxicities resulted in area values of 0.985 and 0.910, respectively (Figure 5), which may be considered outstanding values [25].

Interestingly, the resulting Youden Index values showed that cut-off values respectively of 7.0 and 11.5 mg/L had the highest sensitivity and specificity, although those values were included in a “plateau” ranging from 7.0 up to 15 mg/L (Appendix A). Interestingly, the Youden test identified another peak for hematological toxicities at 7.0 mg/L.

On the contrary, the ROC analysis for gastroenteric toxicities did result in an area value of 0.27 that impeded the identification of a meaningful C_trough_ threshold.

### 3.6. C_trough_ and Toxicities

The existence of a threshold value for C_trough_ has been verified for patients of groups B (hematological adverse events) and C (renal and/or hepatic toxicity) through both the T-test and Fisher’s exact test. In all cases, the obtained *p*-value was less than 0.001.

As already mentioned, although not able to catch all details of DFX pharmacokinetics, the developed pharmacokinetic model is nevertheless significantly robust and reliable enough to provide a good approximation of the C_trough_ value that can play as a dose prediction criterion. Indeed, our pharmacokinetic analysis has demonstrated that DFX C_trough_ values higher than 7.0 mg/L are associated with hepatic and renal toxicities, while C_trough_ values > 11.5 were associated with hematological toxic effects. Therefore, the present POP/PK model may predict DFX dose based on known laboratory parameters.

## 4. Discussion

To the best of our knowledge, the present study has identified a POP/PK model to fit plasma concentrations of DFX in children for the first time. Moreover, the model can investigate every possible correlation between the C_trough_ values and treatment-associated adverse events, especially liver, renal, and hematological toxic effects. Therefore, the model may be a practical tool for DFX dose adaptation to spare patients from severe kidney and liver toxicities from toddlers to adolescents [15,26].

The POP/PK model was developed by adopting the FAMD analysis to search for possible covariates associated with interindividual variability in C_trough_ concentrations. Indeed, the FAMD analysis revealed that two dimensions were responsible for the higher percentage of the variance of DFX pharmacokinetics in our population. The dimensions included the functionality of excretory organs as judged by the presence of plasma creatinine (plus demographic characteristics of patients) in the first dimension, and transaminase plus plasma bilirubin in the second one. Although the methodological rationale for FAMD analysis depended on the accelerated identification of possible model covariates, its results further sustained the appropriate monitoring of liver and kidney functions before and after treatment with DFX because they may have a role in the interindividual variability of plasma concentrations.

The analysis of harvested data and the FAMD results led to the development of a POP/PK model capable of describing the plasma profile of DFX in all patients, including the identification of some variable values that had a significant effect on drug pharmacokinetics. In particular, LBM significantly affected drug bioavailability and k_a_, whereas WT influenced the volume of distribution. Interestingly, parameters of liver functionality (as well as ALT, AST, and direct bilirubin) and renal activity (creatinine) had a significant effect on DFX clearance, and this was not surprising considering that mainly the liver, and also partially the kidney, have a role in drug biotransformation and excretion [15]. On the contrary, the evaluation of further possible covariates and their effects on the final POP/PK model of DFX suggested that those variables related to bone marrow function or peripheral blood counts did not play a role. The patients suffering from hematological toxicities only had C_trough_ values below the threshold of 7.0 mg/L, hence our results suggest that factors other than the sole drug exposure (i.e., AUC) could be responsible for the onset and the severity of neutropenia or anemia. As a matter of fact, several studies focused on the role of pharmacogenetic determinants (i.e., gene polymorphisms) and pharmacokinetic interindividual variability, including genes coding for both enzymes (CYP and UGT isoforms) and transmembrane transporters [20,27,28].

These findings further sustain the careful monitoring of liver and kidney functions in patients treated with DFX because the decrease in the DFX clearance could be greater than 20%. This reduction may expose children to an increased risk of drug-induced toxicities that would have a severe and negative impact on the function and activity of organs already damaged.

To deal with variations in organ functions, the prediction of drug clearance may be more accurate if a bodyweight-based allometric scaling exponent of 0.75 is introduced in the POP/PK model [29]. However, several studies have questioned this strategy, because the exponent value may vary between 0.6 and 1.11 in childhood [30]. For instance, that variability may range from 0.50 up to 1.20 in newborns, toddlers, and children [31]. Due to the large variability of the scaling exponent, the development of the present model included different threshold values of serum creatinine according to the patient’s age, instead of adopting the allometric exponent. Thanks to this feature, the model was capable of analyzing DFX pharmacokinetics while considering the progressive maturation of kidney function over the years.

Therefore, the present POP/PK model did confirm the relationship between drug plasma levels (C_trough_) and treatment-induced toxicities, identifying liver and kidney function laboratory parameters (i.e., plasma bilirubin, creatinine, and transaminases) as the significant covariates to predict DFX pharmacokinetics.

A limitation of the present study could be the number of patients enrolled that are related to the reduced prescription of DFX in patients who received a HSCT, with respect to those affected by beta thalassemia, the other condition associated with IO. Furthermore, the samples were collected according to a TDM protocol available for inpatients but not for outpatients, while smaller patients or those children with suboptimal hemoglobin levels were excluded due to the volume of blood required for the analyses. Despite these potential issues, the developed dose-prediction model appears to be sound and well supported by statistical analyses. Hence, in our opinion, this model may significantly help to spare patients from experiencing renal and hepatic toxicities. Further investigations and large databases could disclose additional criteria to categorize the remaining toxicities.

## 5. Conclusions

In conclusion, to the best of our knowledge, the manuscript presents, for the first time, a POP/PK model to describe the pharmacokinetics of DFX in children, and confirms that C_trough_ values higher than 7.0 and 11.5 mg/L may be associated with liver/renal and hematological toxicities, respectively. Together with a TDM protocol, the model could be introduced into clinical practice to adapt DFX doses in every patient, in order to minimize the occurrence of toxicities while maintaining the therapeutic benefit of the drug.

## Figures and Tables

**Figure 1 pharmaceutics-13-01238-f001:**
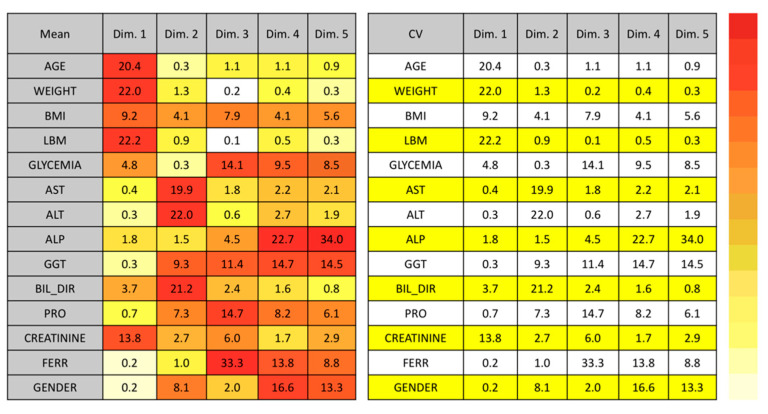
Factor analysis of mixed data (FAMD) results. The darkest red cells on the left correspond to the highest weight of the covariates. Numbers within cells, mean (**left**) and CV (**right**) values. Abbreviations: ALP, alkaline phosphatase; ALT, alanine transaminase; AST, aspartate transaminase; BIL_DIR, direct unconjugated bilirubin; BMI, body mass index; Dim., dimension; FERR, ferritin; GGT, γ-glutamyl transferase; LBM, lean body mass; PRO, total serum proteins.

**Figure 2 pharmaceutics-13-01238-f002:**
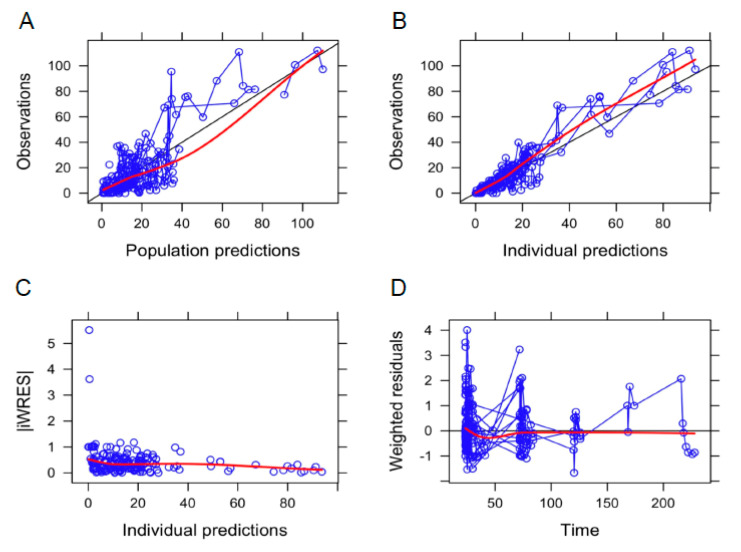
Goodness-of-fit plots for the final pharmacokinetic model. (**A**) Population and (**B**) individual prediction values, (**C**) individual-weighted residuals (|IWRES|) vs. individual predictions, and (**D**) weighted residuals vs. time. Symbols, individual values; red line, LOWESS line.

**Figure 3 pharmaceutics-13-01238-f003:**
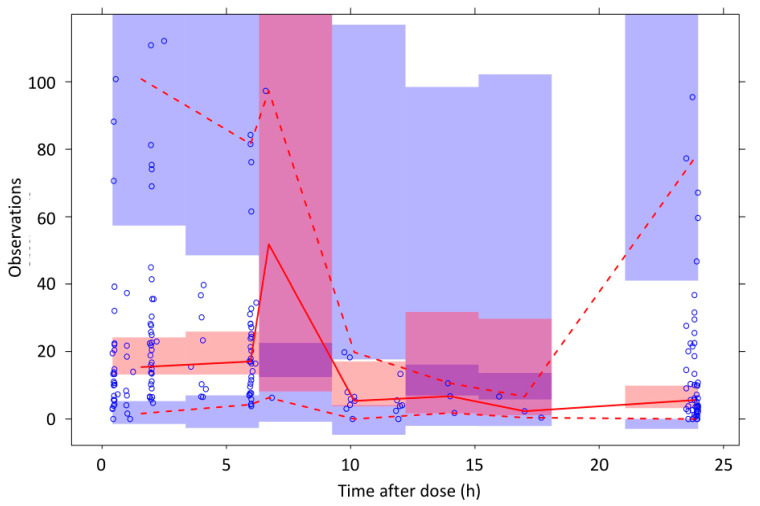
Visual predictive check plot for the final pharmacokinetic model obtained by resampling the original database 2000 times. Symbols, individual measured values of deferasirox plasma concentrations; red lines, median (continuous line) and 95% confidence intervals (dashed lines) of measured values; box, 95% confidence intervals of median (pink) and 95% CIs (blue).

**Figure 4 pharmaceutics-13-01238-f004:**
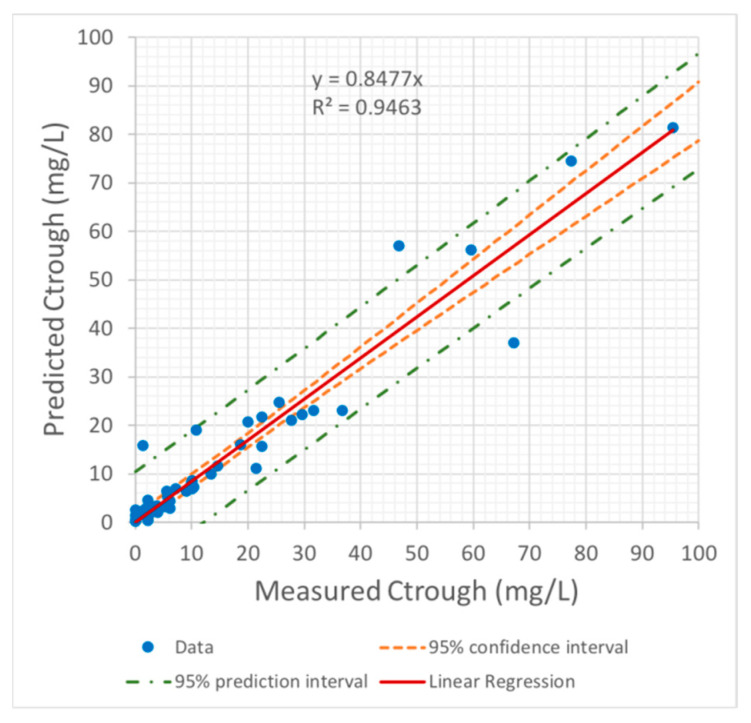
Correlation between predicted and measured C_trough_ values.

**Figure 5 pharmaceutics-13-01238-f005:**
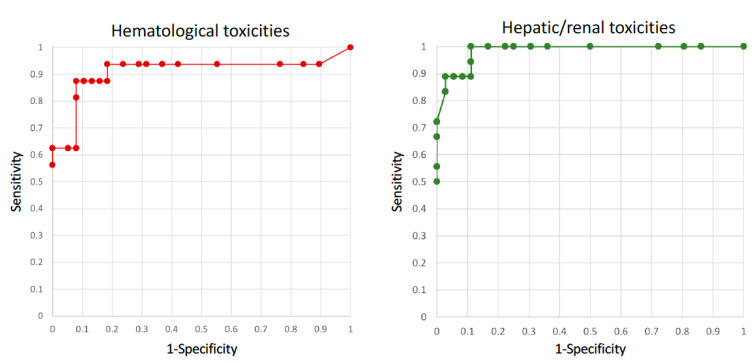
ROC analysis for hematological (**left**) and hepatic/renal toxicities (**right**), with area values of 0.910 and 0.985, respectively.

**Table 1 pharmaceutics-13-01238-t001:** Demographic characteristics of patients enrolled in the present study.

Covariate	Median	IQR ^1^ (%)
Age (years)	9.5	5.3–14.0
Weight (Kg)	30.0	18.6–45.8
BMI (Kg/m^2^)	17.0	14.4–20.0
LBM (Kg)	26.0	17.0–39.2
Glycemia (mg/dL)	91.0	77.3–109.0
AST (U/L)	31.5	22.0–42.0
ALT (U/L)	31.0	24.0–52.8
ALP (U/L)	167.0	114.3–203.0
GGT (U/L)	25.0	17.0–40.0
Direct Bilirubin (mg/dL)	0.2	0.1–0.3
Serum proteins (g/dL)	6.3	5.9–6.7
Creatinine (mg/dL)	0.5	0.3–0.9
Ferritin (ng/mL)	2384.0	1985.8–3690.0

^1^ Abbreviations: IQR, interquartile range; BMI, body mass index; LBM, lean body mass; AST, aspartate transaminase; ALT, alanine transaminase; ALP, alkaline phosphatase; GGT, γ-glutamyl transferase.

**Table 2 pharmaceutics-13-01238-t002:** Creatinine threshold values (*Crea_th_*) adopted in the present study according to patients’ gender and age.

Male		Female	
Age (years)	*Crea_th_* (mg/dL)	Age (years)	*Crea_th_* (mg/dL)
≤2	0.4	≤3	0.4
3–4	0.5	4–5	0.5
5–9	0.6	6–8	0.6
10–11	0.7	9–15	0.7
12–13	0.8	≥16	1.1
14–15	0.9		
≥16	1.3		

**Table 3 pharmaceutics-13-01238-t003:** Results of the final POP/PK model and bootstrap analysis performed in 2000 resampled databases.

			Bootstrap (2000 Samples)	
Parameter	Median Value	RSE ^1^ %	Median	5–95% CI
θ_1_ (L/h)	1.39	11.3	1.35	1.11–1.62
θ_2_ (L/Kg)	1.40	17.1	1.33	0.86–1.81
θ_3_ (1/h)	1.02	19.5	1.02	0.67–1.45
θ_4_ (L/h)	9.16 × 10^−2^	11.1	0.09	0.07–0.11
ω_1_	0.55	11.9	0.54	0.42–0.65
ω_2_	0.48	23.7	0.43	0.19–0.63
ω_3_	0.48	68.5	0.57	0.18–0.92
σ_1, proportional_	0.48	11.0	0.43	0.26–0.51
σ_2, additive_	1.32	27.5	1.70	0.88–8.85
	η-shrinkage		ε-shrinkage	
η_1_	η_2_	η_3_	ε_1_	ε_2_
9.2%	39.8%	60.9%	10.5%	10.5%

^1^ Abbreviations: RSE, root square error; CI, confidence interval.

## Data Availability

The data presented in this study are available on request from the corresponding author. The data are not publicly available due to ethical reasons as per local guidelines.

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
