# Peer review of "Evaluation of Pharmacokinetics and Pharmacodynamics of Deferasirox in Pediatric Patients"

_pharmaceutics, 2021, doi:10.3390/pharmaceutics13081238_

Round 1

Reviewer 1 Report

The topic of the manuscript is interesting. It can be accepted after some minor amendments.

(1) Is age of 17 still considered as pediatric patient?

(2) The sample size of this population pharmacokinetic study is very small. 

(3) Did the authors measured the glucuronidated metabolite as well?

Author Response

Dear Sir/Madam,

On behalf of my coauthors I would like to thank the reviewers for their helpful comments and suggestions. Below, we have listed our replies to referee's queries.

My kindest regards,

Antonello Di Paolo

Reviewer 1

The topic of the manuscript is interesting. It can be accepted after some minor amendments.

(1) Is age of 17 still considered as pediatric patient?

Reply: as per general rule in Italy, the pediatric patient is considered up the age of 18. This suggest that pediatricians have to cope with large interindividual variability in drug response and tolerability.

(2) The sample size of this population pharmacokinetic study is very small.

Reply: as underlined in the discussion section, the limited number of patients may depend on several factors, such as the pathological condition that needed the administration of deferasirox. Indeed, the patients enrolled in the present study were affected by chronic iron overload following an allogenic stem cells transplantation, a condition that is much less frequent than beta thalassemia, the other reason for which deferasirox is prescribed. Furthermore, our analysis was based on blood samples collected in the timeframe 12 - 24 hours after dosing, according to a TDM protocol. Therefore, the samples collection could only be carried out for hospitalized patients. In addition, we could not include in the study smaller patients or patients with suboptimal hemoglobin levels due to the large number of blood samples needed to perform the pharmacokinetic analysis. These issues have been introduced in the Discussion section, page 9 lines 431-436.

(3) Did the authors measure the glucuronidated metabolite as well?

Reply: we did not measure the glucuronide metabolite.

Reviewer 2 Report

The manuscript entitled "Evaluation of pharmacokinetics and pharmacokinetics/pharmacodynamics of deferasirox in paediatric patients" is interesting and discuss an important topic that is worth investigation and attention. 

Minor comments:

  1. Several abbreviations needs to be spelled out in their first appearance
  2. Table 2 and Table 3 labeling and chronological organization in the is confusing
  3. The equations looks cut some letters are not clear.
  4. More details about FAMD in the methods will be useful. Also figure 1 layout and text orientation needs to unified

Author Response

Dear SIr/Madam,

On behalf of my coauthors I would like to thank the reviewers for their helpful comments and suggestions. Below, we have listed our replies to referee's queries.

My kindest regards,

Antonello Di Paolo

Reviewer 2

The manuscript entitled "Evaluation of pharmacokinetics and pharmacokinetics/pharmacodynamics of deferasirox in paediatric patients" is interesting and discuss an important topic that is worth investigation and attention.

Minor comments:

(1) Several abbreviations need to be spelled out in their first appearance

Reply: all of the abbreviations have been carefully checked.

(2) Table 2 and Table 3 labeling and chronological organization in the is confusing.

Reply: the references to Tables 2 and 3 have been checked and the errors have been corrected according to the reviewer’s comment (page 6, libes 276, 279,282, 285).

(3) The equations look cut some letters are not clear.

Reply: the equations have been re-written and corrected following the comment of the reviewer.

(4) More details about FAMD in the methods will be useful. Also figure 1 layout and text orientation needs to be unified.

Reply: FAMD details have been included in the text (lines 138-147), and figure 1 has been reviewed.

Reviewer 3 Report

Manuscript entitled "Evaluation of pharmacokinetics and pharmacokinetics/pharmacodynamics of deferasirox in paediatric patients" by authors Laura Galeotti et al is a clinical study about PK/PD correlation of deferasirox on children. The study was designed well and the results were discussed in details. Results were presented in 3 tables and 5 figures plus additional results in supplemental part. 

Overall this study is good and as authors have mentioned in the manuscript, number of patients is the limitation in this study. Please address the following minor comments

Line#20 please change "every correlation" to "possible correlations"

Line#185: FAMD, please abbreviate 

Authors have addressed mainly the toxicity and discussed less about the pharmacodynamics. 

Author Response

Dear Sir/Madam,

On behalf of my coauthors I would like to thank the reviewers for their helpful comments and suggestions. Below, we have listed our replies to referee's queries.

My kindest regards,

Antonello Di Paolo

Reviewer 3

Manuscript entitled "Evaluation of pharmacokinetics and pharmacokinetics/pharmacodynamics of deferasirox in paediatric patients" by authors Laura Galeotti et al is a clinical study about PK/PD correlation of deferasirox on children. The study was designed well and the results were discussed in details. Results were presented in 3 tables and 5 figures plus additional results in supplemental part. 

Overall this study is good and as authors have mentioned in the manuscript, number of patients is the limitation in this study. Please address the following minor comments

(1) Line#20 please change "every correlation" to "possible correlations"

Reply: the change has been introduced within the text (page 1, line 20).

(2) Line#185: FAMD, please abbreviate

Reply: please note that the paragraph has been moved, while we carefully checked that all of the acronyms have been spelled out at their first occurrence in the text.

(3) Authors have addressed mainly the toxicity and discussed less about the pharmacodynamics.

Reply: we thank the reviewer for the comment. The evaluation of DFX therapeutic effects in relation with drug pharmacokinetics is complicated by the fact that the efficacy in controlling and reducing iron overload becomes evident after a long time. For example, at least three years of deferasirox lead to reversal or stabilization of liver fibrosis in transfusion-dependent thalassemia patients showing evidence of iron overload [Deugnier et al, Gastroenterology 2011;141:1202-1211]. Therefore, the time frame to evaluate every possible correlation between drug pharmacokinetics and efficacy is too long to consider DFX plasma concentrations stable or unaffected by physiological changes during childhood and adolescence.